# Timing between Breast Reconstruction and Oncologic Mastectomy—One Center Experience

**DOI:** 10.3390/medicina56020086

**Published:** 2020-02-20

**Authors:** Adelaida Avino, Laura Răducu, Lăcrămioara Aurelia Brînduşe, Cristian-Radu Jecan, Ioan Lascăr

**Affiliations:** 1Department of Plastic and Reconstructive Surgery, “Prof. Dr. Agrippa Ionescu” Clinical Emergency Hospital, 011356 Bucharest, Romania; adelaida.avino@gmail.com (A.A.); jecan.radu@gmail.com (C.-R.J.); 2Department of Plastic and Reconstructive Surgery, Faculty of Medicine “Carol Davila” University of Medicine and Pharmacy, 020021 Bucharest, Romania; ioan.lascar@gmail.com; 3Department of Public Health and Management, Faculty of Medicine, “Carol Davila” University of Medicine and Pharmacy, 020021 Bucharest, Romania; l.brinduse@yahoo.com; 4Department of Plastic and Reconstructive Surgery, Emergency Clinical Hospital of Bucharest, 014461 Bucharest, Romania

**Keywords:** breast reconstruction, timing, mastectomy, adjuvant therapy, quality of life

## Abstract

*Background and objectives:* Breast cancer is the most common cancer in women. The immunohistochemical profile, but also the stage of the tumor determines the therapeutic management, which varies from conservative surgery to mastectomy associated with chemotherapy, hormonal and biological therapy and/or radiotherapy. Mastectomy remains one of the most radical surgical intervention for women, having great consequences on quality of life, which can be improved by realizing immediate or delayed breast reconstruction. The objective of the study was to evaluate the period of time between the mastectomy and the breast reconstruction. *Material and methods:* We performed a retrospective study on 57 female patients admitted to the Plastic Surgery Department of the Clinical Emergency Hospital “Prof. Dr. Agrippa Ionescu”, Bucharest, Romania. All the patients underwent immediate or delayed breast reconstruction after mastectomy for confirmed breast cancer. Descriptive data analysis was realized with evaluation of type of breast reconstruction considering the staging of the tumor, the invaded lymph nodes, and the necessity of adjuvant chemoradiotherapy. Moreover, the median period between mastectomy and reconstruction was evaluated. *Results:* The immediate breast reconstruction was performed in patients with stage I, in patients with stage II, delayed reconstruction was performed after minimum six months, and the patients with stage III had the breast reconstructed with free flap (50%), 8–43 months post-mastectomy. Radiotherapy determines the type of breast reconstruction, in most of the cases the latissimus dorsi flap was used with implant (22.6%). *Conclusions:* Breast reconstruction is an important step in increasing the quality of life for women who underwent mastectomy after breast cancer. The proper timing for breast reconstruction must be settled by a team formed by the patient, the plastic surgeon, and the oncologist.

## 1. Introduction

Breast cancer is affecting nowadays millions of women worldwide [1]. Unfortunately, the incidence rate has an ascending trend [2]. It is considered to be a multifactorial disease, which is determined by numerous factors, from genetic to environmental ones [1]. The most important step of the treatment plan of breast cancer is surgery. Even if an increasing number of patients are diagnosed in early stages and can benefit of breast-conservatory surgery, total mastectomy is performed in the majority of cases [3]. It has been observed as an increasing tendency for immediate breast reconstruction, mostly in women in whom bilateral mastectomy was performed [4].

The breast represents a part of a woman’s identity, symbolizing femininity, sexuality, beauty, and motherhood. The post-mastectomy scars may have great effect on the body and mind of the patients, especially in young women [5]. Breast reconstruction can be done using autologous tissue or implants. The options of the surgical intervention are established taking into consideration factors such as the type of mastectomy, adjuvant treatments, the body type of each individual, lifestyle, acceptance of possible risks [6]. Another important decisional factor is the patient’s desire. The patient is advised by the plastic surgeon, who explains the options of breast reconstruction. Autologous reconstruction produces a more natural breast, which modifies with the variation of the body mass, being used also after radiotherapy. Implant-based reconstruction is preferred for being quicker, regarding the operating and recovery times [7]. The last method is considered to be the most performed breast reconstruction technique worldwide. In some cases, a tissue expander is utilized prior to the final implant [6]. The symmetry with the opposite breast is one of the final goals. Taking into account the oncological safety, a contralateral breast lift, breast reduction, or augmentation can be recommended to create a balanced result, increasing patient satisfaction [8]. The precise timing of the reconstructive procedure is decided by a multidisciplinary team composed by an oncologist, a radiotherapist, and a plastic surgeon.

In the last 10 years, many studies assessed the satisfaction and aesthetic outcome, in order to highlight the factors for a favorable surgical outcome [7]. Yueh et al.(2010), reported that autologous reconstructions are associated with higher patient satisfaction compared to prosthetic reconstruction [9], a result substantiated by Franco et al.(2018), in a recent publication [10].

The aim of the study was to evaluate the proper timing for the reconstructive procedure for the patients who underwent mastectomy.

## 2. Materials and Methods

We conducted a retrospective study on 57 patients admitted to the Plastic SurgeryDepartment of the Clinical Emergency Hospital “Prof. Dr. Agrippa Ionescu”, Bucharest, Romania, forbreast reconstruction after mastectomy, within a period of 14 months (January 2018–February 2019). The inclusion criteria were: gender (female patients), patients who underwent mastectomy for confirmed breast cancer, immediate or delayed reconstructions. The patients who did not have available pathological data, those who presented local, post-radiotherapy trophic lesions, or metastases were excluded from the study. All data were taken from surgical operating files, medical letters, postoperative records. The preoperative data comprised demographic information, smoking history, but also comorbidities. The histopathological details of the tumor and adjuvant therapies were registered. The different surgical techniques and postoperative complications were analyzed.

Statistical analysis was performed using the SPSS software, version 23.0 and statistical significance was defined as *p* < 0.05. The results were presented as the mean and standard deviations for quantitative variables and as numbers and frequencies for qualitative variables. In order to study the differences among different types of reconstruction techniques, the chi square test was analyzed for qualitative variables, and one-way ANOVA for quantitative variables, respectively. Multivariate analysis was constructed to identify factors associated with the length of time between mastectomy and reconstruction. Covariates for multivariate analysis were selected based on bivariate analyses and included stage of breast cancer, number of affected lymph nodes, and unilateral or bilateral breast cancer.

Local ethical agreement and informed consent of the patient were obtained.

## 3. Results

Fifty-seven cases of immediate or delayed breast reconstructions were performed throughout the period of the study. The average age at primary surgery was 46.74 years (range 37–56 years). From all the patients, 52 were living in an urban area and 16 were smokers. Regarding the breast tumors, in 28 cases ductal carcinoma was discovered, in 21 patients lobular carcinomas, and other types of cancer in eight patients. Twenty-two patients presented stage II cancer, stage III appeared in 18 cases, and stage IV was described for one patient. Axillary lymph node dissection was done in 44 cases, lymph nodes being invaded in 31 individuals (Table 1). Chemotherapy was recommended for 48 patients and radiotherapy for 31 patients. Hormone therapy was indicated in 32 cases. From 57 patients, unilateral breast reconstruction was performed in 46 patients, 10 underwent autologous tissue reconstruction (DIEP free flap). The remaining 48 patients experienced implant-based breast reconstruction: 11 latissimus dorsi flap with implant, 13 expander and implant, five expander Becker, and in 18 cases only implants were used.

There was a statistically significant association between unilateral mastectomy and implant-based reconstruction (*p* = 0.003). The least used was expandable breast implant insertion (10.9%). Moreover, there was an association between bilateral mastectomy and immediate breast reconstruction (Table 2).

In addition, there was a statistically significant association between the stage of the tumor and the type of breast reconstruction (*p* < 0.001).The implant-based breast reconstruction was used in all the patients with stage I, immediate reconstruction being decided in eight cases (50%). In stage II, delayed reconstruction was performed after minimum six months and the most common intervention was latissimus dorsi flap with implant (36.4%). The patients with stage III had the breast reconstructed with free flap (50%), 8–43 months post-mastectomy. The timing between mastectomy and breast reconstruction was determined also by the histological type of the tumor, 17.07 months in case of ductal carcinoma and 7.524 months in case of lobular carcinoma.

Radiotherapy influenced the type of breast reconstruction, a significant correlation between them (*p* = 0.001) being discovered; latissimus dorsi flap with implant was used most frequently (22.6%). Furthermore, the mean duration of hospitalization was strongly associated with the reconstructive intervention (*p* = 0.003).A significant association between postoperative complications and the type of the reconstructive procedure was noticed (*p* = 0.001). The complications were observed in patients with latissimus dorsi flap (46.7%), but also in those with free flap (33.3%). Wound dressings were used to heal the lesions.

There was a significant correlation between chemotherapy and breast reconstruction (*p* = 0.049), but there was not a significant correlation between smoking and type of reconstructive procedure (*p* = 0.077).

## 4. Discussion

Breast cancer is one of the most frequent types of malignancies in women, among lung and colorectal cancer. Thirty percent of all new discovered tumors are breast cancer. Breast reconstruction methods are increasingly performed all over the world. From 2000 to 2016 their number increased with 39%. The most common procedure was the implant-based breast reconstruction. Less than 25% of the patients have immediate reconstruction after mastectomy [11].

In our study, 16 patients were diagnosed with stage I breast cancer and half of them had immediate breast reconstruction with implants (Figure 1). Nipple-sparing mastectomy (NSM) was performed in eight women using a surgical incision in the inframammary fold, five of them had contralateral prophylactic mastectomy, with positive BRCA genes. In three patients, delayed reconstruction with expander was decided. The others had delayed intervention with implants. The reconstructions were performed 3–12 months after mastectomy. None of the patients had radiotherapy. Seven patients had chemotherapy and two of them received also hormonal therapy. One patient presented a minor complication, a seroma.

In patients with stage I breast cancer, the surgical steps have changed over the years, from radical to nipple-sparing mastectomy, increasing the reconstructive and aesthetic results. Even though NSM was considered to have a high oncologic risk, it has been accepted as a reconstructive option in selected patients [12]. This technique is used in younger patients with early breast cancer stages or after prophylactic mastectomy [13].It can be done in one or two surgical steps, but most of the surgeons prefer the one-stage method due to its low overall costs, outstanding outcomes, and low revision rates [12]. Regarding complications, initially, skin and nipple necrosis were considered to be too high, but after being used worldwide, it was emphasized that these two complications have a rate up to 10% and can be treated conservatively, saving the nipple. The most important steps to avoid complications are to make the incisions far from the nipple and to use the inframammary fold, this being demonstrated by Garwood et al. (2009), but also by Colwell et al.(2014) in their studies [13]. In our clinic, this intervention is starting to be used with high frequency due to the fact the patients are presenting in early stages of the disease. Moreover, acellular dermal matrix can be used in immediate reconstruction, to guarantee an optimal placement of the implant [14].

Twenty-two patients had stage II breast cancer and half of them had invasion of three axillary lymph nodes. The reconstruction was indicated after minimum six months after mastectomy, only one patient presented after 26 months. The reconstruction was performed using all five methods described in the study, depending on the surgeon’s decision, but also patient’s desire: One case of free flap, four cases of delayed reconstruction with implants, eight interventions combining autogenous tissue (pedicled latissimus dorsi flap) with implants (Figure 2), and nine were based on expanders, out of which five were with Becker expander. Postoperative complications, wound dehiscences, were observed in patients with latissimus dorsi flap. Special dressings with silver were used to accelerate the healing [15]; fortunately, nowadays there are a lot of modern dressings that aid the cure, reducing pain and local discomfort [16].

Concerning the 18 patients with stage III, the reconstructive procedure was performed 8–43 months post-mastectomy, using free flaps in half of the cases (Figure 3). All the patients presented invasive ductal carcinoma, with at least three invaded lymph nodes. All had chemoradiotherapy. From all 18 patients, the youngest was 30 years old and the eldest 63 years old. In four cases, the breast reconstruction was performed with an expander and in three cases with latissimus dorsi flap and implant. Taking into consideration the complications, we encountered a partial necrosis of one DIEP flap, which was excised early and the defect was covered with a local skin flap.

The patients with stage IV breast cancer do not consider reconstruction after mastectomy as an option, even if new modern therapies against metastasis are available. Their appreciation is that the reconstruction does not have value. However, systemic metastasis is not a contraindication to reconstruct the breast. A proper therapeutic management must be done [17,18]. In our clinic, we had one patient with stage IV breast cancer, with bone metastasis, who had chemoradiotherapy prior to the reconstruction. Her response to the adjuvant therapy was favorable and due to the fact that she was only 42 years old she asked for reconstruction. In her case, the decision was to use latissimus dorsi myocutaneous flap associated with a Becker expander implant. The intervention was performed 32 months post-mastectomy. No complications were observed postoperatively.

Another factor taken into consideration in our study was smoking. Due to the fact that smoking is a contraindication for immediate breast reconstruction, determining skin necrosis and infection [19], none of the smokers had this type of reconstructive procedure.

Another important factor in breast reconstruction is the proper timing after mastectomy. In our study, only 17 patients underwent the reconstruction procedure in the first six months. They had no invaded lymph nodes. In a study made by Lee et al. in 2015, it was demonstrated that the moment of the reconstructive procedure is settled mainly by patient preference, the plastic surgeons taking the responsibility for the best aesthetic result [20], but also by the necessity of postmastectomy radiotherapy [21]. Moreover, the decision of this intervention must be taken after consulting the oncologist [20]. In our study, none of the patients had postmastectomy radiotherapy. According to the stage of breast cancer, immediate reconstruction can be indicated for patients with stage O, I, or IIA. Those with early stage malignancy represent up to 70% of women who go through mastectomy. The new technique of nipple-sparing mastectomy gives an improved outcome and an excellent feedback from the patients. The diagnosis of breast cancer can be overwhelming, especially when the patients are diagnosed in stage IIB or III. Initially, patients focus on adjuvant therapies and lately they take into consideration post-mastectomy reconstruction, so, in these cases the procedure is delayed [21].

In our study, we highlight that it is possible to accurately predict the duration between mastectomy and reconstruction based on the stage of the disease, the number of affected lymph nodes, and the unilateral or bilateral breast involvement. The most important predictor is the stage of the disease (50%), followed by the number of lymph nodes (25%) and the unilateral or bilateral involvement (24%) (Figure 4).

For plastic surgeons, breast reconstruction is a main step in the management of treatment, due to its benefits and impact on the quality of life of patients diagnosed with breast cancer. Worldwide, there are many countries with a low rate of this intervention, even if there are national programs to promote it. Regarding the geographic location, the patients from the rural area are associated with poor chances of having breast reconstruction [22]. In our study, only five patients were from a rural area. In Romania, there is a national program for breast reconstruction after oncologic surgery that covers the prosthesis. The surgical intervention is viewed as a reconstructive procedure, so it is covered by the national health insurance. Unfortunately, breast symmetry is considered to be an aesthetic procedure and it can be performed only in private hospitals.

## 5. Conclusions

Breast cancer is affecting more and more young women. New interdisciplinary protocols are developing for the best therapeutic management, maximizing the outcomes. The timing for breast reconstruction is important, but the decision must be taken after a strong collaboration between the patient, the oncologist, the radiotherapist, and the plastic surgeon.

## Figures and Tables

**Figure 1 medicina-56-00086-f001:**
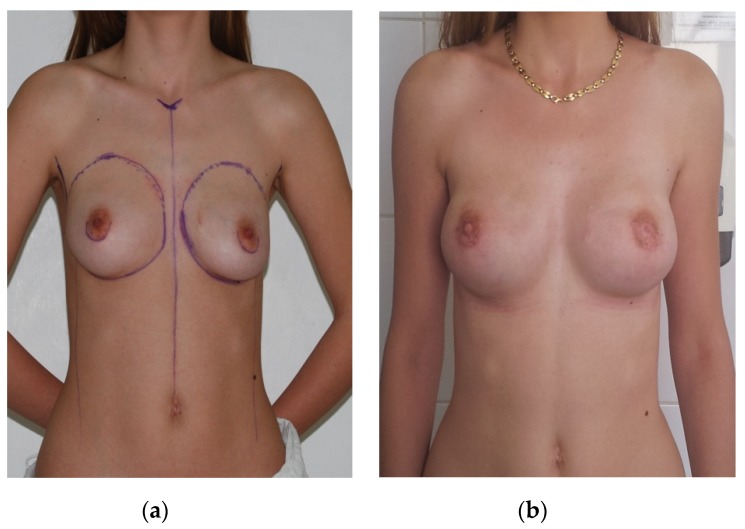
Patient with right breast cancer, stage Iwho underwent bilateral nipple sparing mastectomy with immediate reconstruction. (**a**) Preoperative photo. (**b**)Four months postoperative photo.

**Figure 2 medicina-56-00086-f002:**
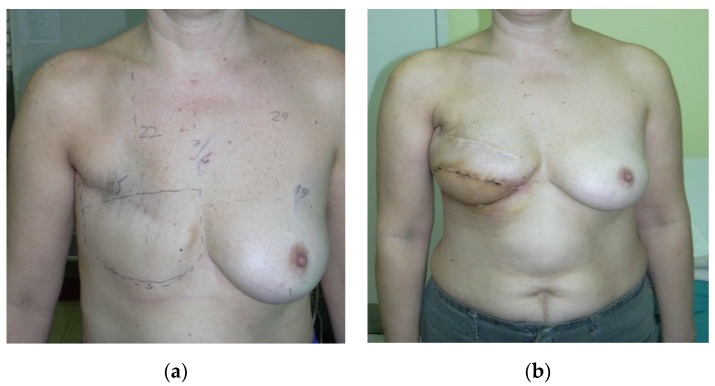
Patient with right breast cancer, stage II who underwent mastectomy. (**a**) Post-mastectomy photo. (**b**) Outcome of delay breast reconstruction with pedicled latissimus dorsi flap and implant shown four months postoperatively.

**Figure 3 medicina-56-00086-f003:**
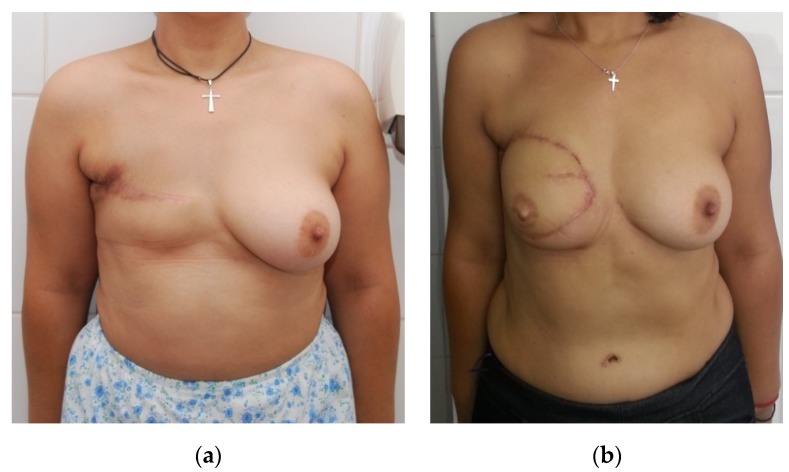
(**a**) Preoperative photo of patient with previous right mastectomy and subsequent radiation treatment. There is inadequate skin surface area to cover a reconstructed right breast. (**b**) Outcome of delayed right breast reconstruction with deep inferior epigastric perforator (DIEP) flap shown eight months postoperatively.

**Figure 4 medicina-56-00086-f004:**
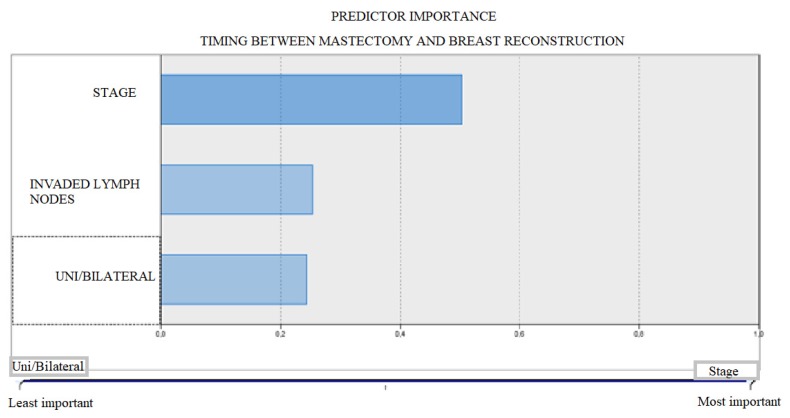
Predictor importance—Timing between mastectomy and breast reconstruction.

**Table 1 medicina-56-00086-t001:** Variables of the study group.

Variable.	N (%)
Axillary Lymphadenectomy	44 (77,2)
Invaded lymph nodes	
0	26 (45,6)
1–3	13 (22,9)
4–9	17 (29,7)
>10	1 (1,8)
Chemotherapy	48 (84,2)
Radiotherapy	31 (54,4)
Hormonal therapy	32 (56,1)
Unilateral	46 (80,7)
Bilateral	11 (19,3)
Type of reconstruction	
Implant	18 (31.6)
Expander-implant	13 (22.8)
Latissimus dorsi flap + Implant	11 (19.3)
Free flap	10 (17.5)
Latissimus dorsi flap + Becker Expander	5 (8.8)
Complications	15 (26,3)
Period of time between mastectomy and reconstruction (median±SD) (months)	11,89 ± 9,11

**Table 2 medicina-56-00086-t002:** Type of reconstruction depending on different variables.

	Type of Reconstruction					
	Implant	Expander + Implant	Latissimus Dorsi + Implant	Free Flap	Latissimus + Expander Becker	*p* Value
	N (%)	N (%)	N (%)	N (%)	N (%)	
Unilateral	9 (19.6)	12 (26.1)	10 (21.7)	10 (21.7)	5 (10.9)	0.003
Bilateral	9 (81.8)	1 (9.1)	1 (9.1)	0 (0.0)	0 (0.0)	
Complications	1 (6.7)	0 (0.0)	7 (46.7)	5 (33.3)	2 (13.3)	0.001
Stage						<0.001
I	13 (81.2)	3 (18.8)	0 (0.0)	0 (0.0)	0 (0.0)	
II	4 (18.2)	6 (27.3)	8 (36.4)	1 (4.5)	3 (13.6)	
III	1 (5.6)	4 (22.2)	3 (16.7)	9 (50.0)	1 (5.6)	
IV	0 (0.0)	0 (0.0)	0 (0.0)	0 (0.0)	1 (100.0)	
Radiotherapy	4 (12.9)	6 (19.4)	7 (22.6)	1 (32.3)	4 (12.9)	0.001
Chemotherapy	12 (25.0)	10 (20.8)	11 (22.9)	10 (20.8)	5 (10.4)	0.049
Smoking	4 (25.0)	4 (25.0)	5 (31.2)	0 (0.0)	3 (18.8)	0.077
Hospital days (mean ± SD)	8.6 ± 4.6	10.1 ± 3.6	15.5 ± 5.6	13.7 ± 4.9	12.6 ± 4.2	0.003

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
