# Peer review of "Timing between Breast Reconstruction and Oncologic Mastectomy—One Center Experience"

_medicina, 2020, doi:10.3390/medicina56020086_

Round 1
Reviewer 1 Report
The article submitted by the authors highlights the importance of timing breast reconstruction after mastectomy for confirmed breast cancer treatment, which reflects important effects on quality of life. The article shows a good English and the bibliography is coherent and recent. However, I have some comments:
Point 1: the introduction of the manuscript should be more specific in clarifying the aim of this study. In lines 73-74 (page 2), the authors write that “The aim of the study was to evaluate the period of time between the mastectomy and the breast reconstruction”. Why do the authors evaluate it? What is their final aim? In my opinion, it could help the manuscript to be more fluid.
Point 2: In lines 135-136 (page 4) the authors write “There was no significant correlation between chemotherapy and breast reconstruction (p=0.049)”. If statistical significance was defined as p<0.05, why do the authors consider p=0.049 not statistically significant?
Author Response
- We wanted to evaluate the timing of the breast reconstructive procedure after mastectomy in one medical center in Romania.
- I have mistaken: the correct form is :
There was a significant correlation between chemotherapy and breast reconstruction (p=0.049), but there was not a significant correlation between smoking and type of reconstructive procedure (p=0.077).
Reviewer 2 Report
The first phrases from the Discussion section need to add references. Minor English spelling mistakes should be checked. A few words appear "sticked", especially in the References section.
Figure 4: please add numbers (percentages) to the graph.
Author Response
I have modify the refereces.
The figure 4 is from the SPSS software. I could not modify from my computer. If is ok if I left it like this?
Thanks a lot from the understanding.